# Mitophagy and the Brain

**DOI:** 10.3390/ijms21249661

**Published:** 2020-12-18

**Authors:** Natalie S. Swerdlow, Heather M. Wilkins

**Affiliations:** 1University of Kansas Alzheimer’s Disease Center, University of Kansas, Kansas City, KS 66160, USA; natswerd@icloud.com; 2Department of Neurology, University of Kansas Medical Center, Kansas City, KS 66160, USA; 3Department of Biochemistry and Molecular Biology, University of Kansas Medical Center, Kansas City, KS 66160, USA

**Keywords:** mitochondria, Alzheimer’s Disease, mitophagy, neurodegeneration, aging

## Abstract

Stress mechanisms have long been associated with neuronal loss and neurodegenerative diseases. The origin of cell stress and neuronal loss likely stems from multiple pathways. These include (but are not limited to) bioenergetic failure, neuroinflammation, and loss of proteostasis. Cells have adapted compensatory mechanisms to overcome stress and circumvent death. One mechanism is mitophagy. Mitophagy is a form of macroautophagy, were mitochondria and their contents are ubiquitinated, engulfed, and removed through lysosome degradation. Recent studies have implicated mitophagy dysregulation in several neurodegenerative diseases and clinical trials are underway which target mitophagy pathways. Here we review mitophagy pathways, the role of mitophagy in neurodegeneration, potential therapeutics, and the need for further study.

## 1. Introduction

Mitochondria are essential organelles that regulate energy homeostasis, cell signaling, and cell death [1,2,3]. During threatened cell death or nutrient starvation, mitochondria can be degraded and recycled through mitophagy. Mitophagy is a specific form of autophagy, a process where cell contents are degraded and recycled. In a broad sense, mitophagy involves tagging mitochondria for removal, engulfment of the organelle by an autophagosome, and degradation in a lysosome. There are several pathways which control, initiate, and facilitate mitophagy. The many facets of mitochondrial function contribute to mitophagy pathways.

Mitochondria coordinate and balance energy production through beta oxidation, the citric acid cycle (TCA cycle), and oxidative phosphorylation at the electron transport chain (ETC). Beta oxidation is a catabolic pathway where free fatty acids are converted to acetyl coA, which enter the TCA cycle and ultimately oxidative phosphorylation. In the TCA cycle, either pyruvate (from glycolysis) or acetyl coA are oxidized to generate the high energy electron carriers, NADH and FADH_2_. NADH and FADH_2_ enter the ETC at complex I and complex II, respectively. These high energy electron carriers undergo oxidation/reduction reactions in the ETC in order to pump protons into the matrix. These protons ultimately power ATP synthase (or Complex V) for the generation of ATP from ADP. These bioenergetic reactions maintain the mitochondrial electrochemical gradient, or mitochondrial membrane potential. Mitochondrial membrane potential is the main signal which either inhibits or initiates mitophagy.

Mitochondria are double membrane organelles. The outer mitochondrial membrane is imperative for mitophagy function [4,5,6,7]. Transport and signaling proteins localize to the outer mitochondrial membrane to facilitate protein and metabolite import. As discussed in more detail below, aggregation-prone proteins are known to block these import channels on the outer mitochondrial membrane in some neurodegenerative diseases. This could lead to the disruption of mitophagy processes and requires more research efforts to understand. The inner membrane space of mitochondria houses enzymes and allows for proton storage and protein folding. The mitochondrial inner membrane contains the ETC and ATP synthase enzymes and the mitochondrial matrix stores enzymes for the TCA cycle, mitochondrial DNA (mtDNA), and other crucial enzymes for protein folding and maintenance of pH gradients. For more detailed analysis of mitochondrial localized proteins MitoCarta3.0 was recently published [8].

Synaptic loss is strongly correlated with cognitive deficits and motor dysfunction [9,10,11,12]. Mitochondria are essential for synaptic function and neurotransmitter synthesis, release, and uptake [13,14,15]. Accumulation of damaged mitochondria could lead to synaptic dysfunction and neurodegeneration. Mitophagy may play a role in ensuring synaptic mitochondrial integrity by degrading damaged mitochondria.

Mitochondria evolved from a prokaryotic endosymbiont. As such, mitochondria share characteristics with bacteria including a double membrane, circular DNA, formyl-methionine amino acids, and cardiolipin [1]. In certain contexts failure of mitophagy pathways could lead to the release of mitochondrial components into the extracellular space, activation of a damage-associated molecular response (DAMP), and inflammation [1,5]. Mitochondria are also master regulators of cell death pathways (such as apoptosis and necrosis) [4,5,16]. As a by-product of the respiratory chain function superoxide radicals are produced. These free radicals generate multiple species of reactive oxygen species (ROS) and reactive nitrogen species (RNS). During periods of mitochondrial dysfunction and failure of mitochondrial quality control mechanisms (such as mitophagy) ROS/RNS can induce damage to cellular macromolecules and necrotic cell death [4,5,16]. Proper control and coordination of mitophagy pathways are crucial to prevent cell death and inflammation.

Mitophagy is imperative for glial cell function. Signaling between microglia, astrocytes, and neurons are modulated by mitophagy pathways. Novel data show that transcellular mitophagy pathways occur within the brain. Transcellular mitophagy is a process by which cells release mitochondria for engulfment and mitophagy in surrounding cell types. Dysregulation of this process can lead to neuroinflammation and loss of proteostasis [17,18,19].

Disruption of mitophagy is observed with aging and in many neurodegenerative diseases. Recent advances have described novel mechanisms of mitophagy within the central nervous system (CNS). Here, we will review current knowledge of mitophagy regulation, its role in neurodegenerative disease, and therapeutic potential.

For this review article, we used PubMed, clinicaltrials.gov, and Google Scholar to identify studies related to mitophagy, AD, PD, ALS, and MS. We used search terms including mitophagy, mitophagy and neurodegeneration, mitophagy and AD, mitophagy and PD, mitophagy and MS, mitophagy and ALS, mitochondria and neurodegeneration, and autophagy.

## 2. Mitophagy: Autophagy for Mitochondria

Autophagosomes can be derived from membranes of endoplasmic reticulum (ER), Golgi, mitochondria, or plasma membrane [20,21,22,23,24]. The biogenesis of autophagosomes involves the formation of the isolation membrane (IM), elongation and maturation, closure, and then fusion with the lysosome to form the autolysosome. The first step in autophagosome biogenesis is the activation of the Unc-51-like kinase 1 complex (ULK1; pre-initiation complex), which contains ULK1, autophagy-related proteins 13 and 101 (Atg13, Atg101), and focal adhesion kinase family interacting partner 200 (FIP200) [25,26,27,28]. This complex recruits the class III phosphatidylinositide 3-kinase (PI3K) Vps34 complex (Beclin1, autophagy-related protein 14 (Atg14), autophagy and beclin 1 regulator 1 (Ambra1), and vascular protein sorting 34 and 15 (Vps34 and Vps15)) to produce phosphatidylinositol 3-phosphate (PI3P), also called the initiation complex [25,27,29,30]. PI3P binding proteins, FYVE domain containing proteins (DFCP1), and WD repeat protein interacting with phosphoinositide (WIPIs) localize to the IM [26]. All of this culminates in the formation of the omegasome and IM. Autophagy-related proteins 12, 5, and 16 (Atg12, Atg5, and Atg16) and LC3-phosphatidylethanolamine (PE) facilitate the elongation and closure of the IM, following fusion with the lysosome [25,26,31].

Mitochondrial quality control mechanisms include mitochondrial chaperones, the mitochondrial unfolded protein response (UPR^mt^), degradation of mitochondrial proteins in the cytoplasm via the proteasome (p97, 26S proteasome), removal of damaged proteins via mitochondrial derived vesicles (MDVs), and mitophagy (Figure 1) [32]. Misfolded mitochondrial proteins can be refolded by mitochondrial chaperone proteins (heat shock proteins 22, 60, and 70) or cleaved/degraded by mitochondrial proteases Lon and Clp through the UPR^mt^ [33,34,35,36,37]. Damaged mitochondrial proteins can also be targeted specifically for proteasome degradation by p97 through the proteasome [38]. MDVs bud off mitochondria after engulfing damaged mitochondrial macromolecules. MDVs are associated with impaired mitochondrial import channels. These MDVs are either degraded by lysosomes, peroxisomes, or exocytosed [39,40]. Mitophagy (reviewed extensively below) functions to remove damaged mitochondria with the goal of preventing cell death.

### 2.1. Non-Receptor Mediated Mitophagy

Non-receptor mediated mitophagy (Figure 2) or classical mitophagy involves PTEN-induced kinase 1 (PINK1) and Parkin. PINK1 is normally degraded in the inner mitochondrial membrane (IMM) by PARL (presenilin-associated rhomboid-like protease) [41,42]. During mitophagy induction, PINK1 becomes active and accumulates on the outer mitochondrial membrane (OMM), where it recruits Parkin and ubiquitin through phosphorylation [43,44,45,46,47,48,49,50,51]. Parkin accumulates and polyubiquitinates mitochondrial proteins triggering proteasomal degradation. Parkin ubiquitination of OMM proteins leads to more PINK1 activity and phosphorylation of substrates, including the recruitment of additional Parkin proteins at the OMM, creating a positive feedback loop. OMM proteins mitofusins 1 and 2 (MFN1/2), voltage-dependent anion channel 1 (VDAC1), and translocase of the outer mitochondrial membrane 20 (TOM20) are ubiquitinated by Parkin [44,46,47,48,49,50,51,52,53,54,55]. Ubiquitination of MFN1/2 blocks mitochondrial fusion allowing for the isolation of damaged mitochondria and smaller mitochondria facilitate autophagosome targeting [6,32,49,56].

Autophagy adaptor proteins are recruited to the OMM by ubiquitinated proteins. These adaptors include neighbor BRCA1 gene (NBR1), nuclear dot protein 52 (NDP52), optineurin (OPTN), sequestosome-1 (SQSTM1/p62), and Tax1-binding protein (TAX1BP1) [23,52,57,58,59,60,61,62,63,64]. Adaptor proteins recruit and interact with autophagosome proteins; gamma-aminobutyric acid receptor-associated protein (GABARAP) or microtubule-associated protein 1A/1B-light chain 3 (LC3) to mediate the formation of the mitophagosome and lysosomal fusion/degradation. These adaptor proteins interact with LC3 and GABARAP through LC3 interacting regions (LIR) motifs (W/F/YxxL/I) [6,7,23,32]. The importance of different adaptor proteins has been up for debate and often their involvement in mitophagy is context-dependent. The same is true for Parkin, as there are some pathways of mitophagy which are Parkin-independent and these are discussed below [23,44].

### 2.2. Receptor Mediated Mitophagy

Receptor mediated mitophagy (Figure 2) is driven by mitochondrial receptor proteins which contain LIR motifs (W/F/YxxL/I) [23,50]. Outer mitochondrial membrane proteins, autophagy and Beclin 1 regulator 1 (AMBRA1), Bcl-2 interacting partner 3 (BNIP3), FUN14 domain-containing protein 1 (FUNDC1), Nip3-like protein X (NIX); and inner mitochondrial membrane proteins cardiolipin and prohibitin 2 (PHB2) are the most studied receptors which mediate mitophagy [23,50,65,66,67,68,69,70].

Receptor mediated mitophagy is activated under specific conditions. For example, during hypoxia BNIP3 and NIX transcription are activated by hypoxia inducible factor 1 alpha (HIF1α) [70,71,72,73]. BNIP3 and NIX activity are regulated by phosphorylation where increased phosphorylation increases their binding affinity for LC3 [74,75]. Hypoxia also promotes FUNDC1 binding to LC3 through dephosphorylation via phosphoglycerate mutase family member 5 phosphatase (PGAM5) [65,67,76,77]. Conversely, FUNDC1 phosphorylation by ULK1 is also a mitophagy activating event. Ultimately ubiquitination of FUNDC1 by E3 ubiquitin protein ligase 5 (UBR5) promotes lysosomal degradation of mitochondria [65,67,78]. The inner mitochondrial membrane receptors, PHB2 and cardiolipin have been shown to interact the LC3, especially during times of mitochondrial permeabilization [79,80]. AMBRA1 is sequestered and inhibited by B-cell lymphoma protein 2 (Bcl-2) on the outer mitochondrial membrane but upon mitophagy activated AMBRA1 binds LC3 in a Parkin-dependent or -independent manner [81].

Receptor mediated mitophagy culminates in the elongation of and closure of phagophore membranes, resulting in engulfment of the mitochondria. The elongation and closure of the phagophore is driven by the mitochondrial receptor binding LCR and/or GABARAP, leading to closure of the phagophore by GABARAP [23,50]. Lastly, the autophagosome fuses with a lysosome for degradation.

### 2.3. Mitoptosis

Separate from mitophagy pathways, damaged mitochondria can be partitioned and removed through mitoptosis. This phenomenon was first proposed in 1992 [82]. Mitoptosis has several proposed definitions. One possible process of mitoptosis occurs when damaged mitochondria gather around the nucleus, are selectively partitioned into lipid membranes and extruded from the cell [83]. A separate definition is when mitochondria undergo condensation with swelling and fragmentation of cristae. This leads to the bursting of the outer mitochondrial membrane and fragmented cristae are extruded into the cytoplasm. In other forms of mitoptosis the outer mitochondrial membrane can remain intact and the cristae deteriorate through refraction and coalescence [6,83,84,85,86,87]. The main benefit of mitoptosis is to prevent opening of the mitochondrial permeability transition pore (MPTP) and apoptosis. Overall, the method of which cells dispose of damaged mitochondria or part of mitochondria can vary and requires further study. Cell-specific pathways which evolved to provide the least devastating consequences based on cell and tissue function likely exist.

Mitoptosis does not require extramitochondrial signaling or protein complexes according to current knowledge. Some evidence suggests PINK1 and Parkin are involved, but no consensus currently exists [6,83,84,85,86,87,88]. Situations which likely activate mitoptosis include mitochondrial membrane depolarization, increased ROS production, and degradation of mtDNA.

### 2.4. Transcellular Mitophagy

Recent studies have described exocytosis of mitochondria from cells followed by endocytosis or phagocytosis of these extracellular mitochondria. In the brain neurons release mitochondria at synapses, and these extracellular mitochondria were taken up by glial cells for phagocytosis [17,18]. This phenomenon will be referred to as transcellular mitophagy here (Figure 3). A recent study showed that retinal ganglion axons shed mitochondria, which were then then degraded by adjacent astrocytes in mice [18]. Mitochondria might undergo degradation in the axoplasm (cytoplasm of the nerve axon) with the assistance of axonal lysosomes [17]. Essentially instead of being degraded in the soma transcellular mitophagy describes the fact that axonal mitochondria are instead enclosed by axoplasmic membranes that are shed and degraded by neighboring cells.

The notion that transcellular mitophagy occurs is logical given that it is not energetically favorable for neurons to bring mitochondria from dendrites back to the cell body for mitophagy. The process of transcellular mitophagy requires more study and understanding at the basic level and with regards to disease.

In addition to transcellular mitophagy, the observation of glial cells transferring mitochondria to neurons has been documented. In stroke models, glial cells transfer mitochondria to neurons as a likely means to protect neurons from energy stress and hypoxia [89,90]. The specific pathways which facilitate mitochondrial transfer between cells are unknown. However, some studies suggest that in astrocytes, glial acidic fibrillary protein (GFAP), and in neurons, uncoupling protein 2 (UCP2) may play a role [91,92].

In mesenchymal stem cells, connexins (particularly connexin 43) oligomerize to form Gap junctions. These Gap junctions may facilitate the formation of tunneling nanotubules, which allow the exchange of cellular contents, such as mitochondria [93]. Other studies suggest Miro1, a protein which connects cytoskeletal motor proteins to mitochondria is involved in mitochondrial transfer between cells [94]. Finally, S100A4 guides tunneling nanotubule growth [95]. The phenomenon of mitochondrial transfer between cells has been documented in a wide variety of model and tissue types [17,89,90,92,93,94,95,96,97,98,99,100,101]. An important question remains regarding what initiates mitochondrial transfer and the specific mechanisms which control this process.

## 3. Mitophagy in Aging and Neurodegeneration

Changes in mitophagy flux, signaling, and mitochondrial function are observed with aging and in neurodegenerative diseases. It is imperative to understand the role mitophagy could contribute to brain aging and neurodegeneration. We discuss these implications below.

### 3.1. Aging

Aging is associated with a loss of proteostasis, mitochondrial dysfunction, genome instability, inflammation, changes to redox balance, and metabolic deficits [32]. Reduced autophagy and mitophagy are observed in models of aging [32,102].

Mitochondrial homeostasis and function are altered in aging. However, the exact mechanisms and findings are not consistent across models and studies. Brain cytochrome oxidase (COX; complex IV) and complex I activity are reduced while ROS production and oxidized proteins are increased in aged rats [103,104]. Calcium homeostasis is changed in aged rat brain mitochondria and synaptosomes; neither were able to take up calcium at a rate equivalent to young rats [105]. Aged mice have altered proteomic expression of glycolytic, TCA, and oxidative phosphorylation pathways in the brain but mitochondrial function is unchanged [106]. This suggests a compensatory mechanism during aging.

Altered brain mitochondrial morphology is observed in aged rats and monkeys [107,108]. Other findings suggest a change to mtDNA epigenetic markers and increased mtDNA deletions in aged mouse brain [109,110]. Oxidative damage to brain mtDNA is related to reduced lifespan across numerous species (birds and mammals) [111,112]. mtDNA in the aged human brain shows increased somatic mutation burden and oxidative damage [113,114,115].

In multiple organisms, mitophagy is associated with longevity and lifespan. *Pink1* knockout causes a shorter lifespan, while *Parkin* overexpression in neurons increases lifespan in *D. melanogaster* [45,116,117]. *C. elegans* models exposed to mild mitochondrial stress and upregulated mitophagy have extended lifespan [118,119]. Urolithin A (UA), tomatidine, and catechinic acid induce mitophagy and increase the lifespan of *C. elegans* models [120,121,122]. In an aging mouse model, stimulation of mitophagy with NAD^+^ prolongs lifespan [123].

Mitochondrial quality control emerges as a central theme in most neurodegenerative diseases, including Alzheimer’ Disease (AD), Parkinson’s Disease (PD, Multiple Sclerosis (MS), and Amyotrophic Lateral Sclerosis (ALS). Mitophagy stimulation has shown positive effects in models of these diseases.

### 3.2. Alzheimer’s Disease

AD is the most common form of dementia diagnosed upon autopsy with neuropathological examination [124,125]. The pathological hallmarks which lead to AD diagnosis postmortem are considerable Aβ plaques and tau tangles throughout the brain [124,125,126,127]. Recent advances in neuroimaging have allowed the determination of Aβ plaque and tau tangle load in living subjects, showing these proteins accumulate in the brain decades before clinical signs of cognitive decline [126,128,129,130].

One of the earlier observations in AD subjects was reduced glucose uptake/utilization in the brain via fluorodeoxyglucose (FDG)-positron emission tomography (PET) [128,129,130,131,132,133]. Accumulation of evidence supports an overall metabolic deficit in AD subjects both within the brain and systemically [134]. The mitochondrial ETC enzyme, COX (or complex IV), has reduced Vmax in the AD brain, fibroblasts, and blood samples [132,135,136,137,138,139,140,141,142,143,144,145,146]. AD-like changes can be transferred to other cell types when AD patient mitochondria (mtDNA) are transferred [139,147,148,149,150,151]. This process of creating cytoplasmic hybrid cells (cybrids) allows for the determination of the contribution of mtDNA on disease and cell physiological processes [150].

Mitochondria in AD autopsy brain samples have fragmented cristae and vary widely in size compared to age-matched non-demented brain samples [152]. Mitochondria within dendritic spines and presynaptic terminals show the most fragmented and disorganized cristae. Alterations to mitochondrial ultrastructure were observed in areas of the brain with and without Aβ and tau pathology (cerebellar cortex, hypothalamus, cerebellum, and visual cortex). In addition to altered mitochondrial morphology, presynaptic terminals had reduced synaptic vesicles and fragmentation of golgi cisternae was observed [152].

mtDNA inheritance confers risk to AD. Studies show that offspring of maternal AD subjects have a higher risk of AD diagnosis than offspring of paternal AD subjects. Nearly all mitochondria are inherited maternally. Offspring from maternal AD subjects show metabolic and neuroimaging changes earlier in life than offspring of paternal AD subjects [129,153,154,155,156]. Furthermore, inherited mtDNA haplogroups are associated with both increased and decreased AD risk [157,158,159,160,161,162,163,164]. These inherited mtDNA haplogroups also interact with the nuclear DNA encoded risk factor, ApoE (apolipoprotein E) to influence AD risk [159,164,165]. Thus, it is important to understand the role of mitochondria, mitophagy, and metabolism in AD.

AD mouse models have disrupted mitophagy [166]. This is observed in tau transgenic mice and AD postmortem human brains with accumulated tau aggregates [167]. Mutant Amyloid Precursor Protein (APP; Swedish mutant) mice show increased mitochondrial fission proteins and decreased mitochondrial fusion and mitophagy protein expression in hippocampal neurons [168]. APPsw/PS1dE9 transgenic mice show increased LC3, PINK1, and Parkin expression [169]. Cortical neurons derived from AD transgenic mice (J20; Swedish and Indiana APP mutations) also show increased mitophagy protein expression with depolarized mitochondrial membrane potential [170].

In AD postmortem brain samples, accumulation of damaged mitochondria and autophagosome vacuoles is observed [171,172,173]. The UPR^mt^ pathway is upregulated at the gene level, with reduced proteasomal activity through the 26S proteasome. Parkin, SQSTM1/p62, and LCR mitochondrial localization are increased [172,173,174,175,176]. Mitophagy pathways are altered in human postmortem AD brain. Cytosolic Parkin is depleted, and lysosomal deficits are observed. Impaired Parkin recruitment to mitochondria is possibly caused by tau-mediated sequestration of Parkin in the cytosol [170,177]. Defects in the activation of ULK1 and TBK1 lead to impaired mitophagy [177]. In AD, mitophagy increases or decreases depending on the part of the cell observed. Although it is increased in lysosomes, other parts of the cell fail at completing mitophagy.

Mechanisms of altered mitophagy and mitochondrial function warrant further study in AD, specifically given the strong association of mitochondria and mitophagy with synapse health and function.

### 3.3. Parkinson’s Disease

PD is a neurodegenerative disease with both cognitive and neuromuscular changes. Motor deficits, tremors, rigidity, bradykinesia, dyssomnia, and depression are clinical hallmarks of PD. In some cases, PD can cause cognitive impairment. Within the brain, PD causes degeneration of dopaminergic neurons in the substantia nigra with Lewy body accumulation (composed of aggregating α-synuclein) [178,179,180].

Mitochondrial dysfunction is observed in PD. A complex I deficiency is noted in brain tissue (substantia nigra) but not in skeletal muscle [181,182]. The complex I deficiency might be brain-specific, but some studies suggest deficits in platelets of PD subjects [183,184]. These findings are dependent on the methodology used. Mitochondrial dysfunction is also observed in cybrid cell lines derived from transfer of PD subject mtDNA, suggesting mtDNA may play a role [150,185,186,187]. Furthermore, PD patients have increased rates of mtDNA deletion in the substantia nigra [188,189].

Familial forms of PD are caused by mutations in genes involved in mitophagy. Mutations in *PARK6* (encodes for PINK1), *PARK2* (encodes for Parkin), *PARK1/4* (α-synuclein), *PARK7* (DJ1), *PARK8* (LRRK2), *PARK17* (Vsp35), and *PARK9* (ATP13A2) genes are linked to familial PD [177,190,191,192,193,194,195]. The role of PINK1, Parkin, and Vsp35 in mitophagy are well known and reviewed above. DJ1 and α-synuclein have been shown to modulate mitophagy either through direct interactions with PINK1 and Parkin or by causing mitochondrial fragmentation. Loss of function of LRRK2 and ATP13A2 have been shown to impair mitochondrial turnover.

Despite these genetic studies most PD cases are sporadic with no known genetic cause. Inhibition of complex I function with rotenone (a pesticide) or 1-methyl-4-phenyl-1,2,3,6-tetrahydropyridine (MPTP) induces PD in rodent and non-human primates [196,197,198,199,200]. Dopaminergic neurons form many synapses (up to one million per neuron), and thus, have a high bioenergetic demand to maintain these unmyelinated synapses [9,201]. In human postmortem brain, mitophagy markers (phosphorylated S65 ubiquitin) increase across age and with PD diagnosis [202]. This mitophagy marker also associated with Lewy Bodies, showing an increase of mitophagy in early PD stages and a decrease in late PD stages [202]. α-synuclein is also associated with an increase in Miro expression in postmortem human brain tissue, human neurons, and fly models of PD [203]. Reducing Miro in the human neuronal and fly models rescued neurodegeneration and mitophagy [203].

Overall, PD patients and disease models consistently show mitochondrial and mitophagy dysfunction.

### 3.4. Multiple Sclerosis

MS is a neurodegenerative disease marked by an autoimmune response against the myelin sheath. Demyelination of white matter is caused by autoreactive T cells which target the myelin sheath within the central nervous system. This demyelination leads to a secondary loss of neuronal axons and neurodegeneration. MS has no known genetic cause and occurs in young adults with a higher incidence in females (3:1 ratio female to male) [204,205]. Most MS subjects (85%) have a relapse–remission disease course, which includes periods of demyelination followed by neurological recovery. Eventually a secondary progressive course occurs with few remission periods [204,205]. A smaller subset of MS patients (10–15%) show continued progression known as primary progressive MS.

MS pathology begins with the formation of a lesion with acute inflammation, which transitions to a state of chronic inflammation followed by neurodegeneration. Chronic inflammation is believed to allow penetration of the blood brain barrier by activated T cells directed against the myelin sheath. The inflammation observed in MS shows activation of both innate and adaptive immunity pathways [204,205]. In addition to demyelination, loss of oligodendrocytes is observed. Myelin autoreactive T-cells can induce experimental autoimmune encephalomyelitis (EAE) in animal models and human MS subject genome wide association studies (GWAS) have a high representation of immune genes related to T cell differentiation [204,205]. Gray matter lesions and brain atrophy are present before clinical MS onset. These findings highlight the lack of understanding of what ultimately initiates the autoimmune reaction in MS.

Mitochondria and mitophagy are critical for immune signaling. ROS signals from mitochondria are known to modulate inflammatory responses. ROS signals in MS damage myelin and the blood brain barrier, further exacerbating disease. Oxidation of phospholipids and DNA damage are observed during periods of chronic inflammation with disruption of neuronal axons. Chronic inflammation induces damage to macromolecules including mtDNA, ETC protein, and lipids [204]. Systemic mitochondrial changes are observed in MS. Peripheral lymphocytes (mostly T cells) show increased mitochondrial superoxide production, decreased ETC protein expression, increased lactate, and decreased antioxidant capacity in human MS subjects [206]. These findings suggest a mitochondrial and bioenergetic deficit in MS.

Of interest, MS subjects show decreased expression of COX5B [204]. Active lesions from MS subjects show reductions in COX and its catalytic component COX-I; decreased expression was explicitly observed in oligodendrocytes, astrocytes, and neuronal axons [207]. A separate study showed reduced COX activity in lesions from MS patients; they also observed correlation of this endpoint with neurofilament protein (SMI32) expression and with macrophage/microglial density [208]. An axon-specific protein, syntaphilin, which functions as a mitochondrial docking protein, was increased in chronic lesions [208]. Inactive lesion areas in the same MS subjects had elevated COX activity and increased mitochondrial mass [208]. Gene expression analysis of cortex tissue from MS subjects show overall reductions in nuclear encoded mitochondrial genes and ETC complex expression specifically in neurons [209]. Overall, mitochondrial deficits are observed in MS and the role of these deficits requires further study.

Autophagy and mitophagy pathways are altered in MS human subjects and animal models, which mimic MS pathology. Atg5 modulates T cell survival and its expression correlates with clinical disability in mouse models of EAE. In MS brain samples, encephalitogenic T cells appear to be the major source of Atg5 expression and systemic T cells in human MS subjects showed increased Atg5 expression [210]. In MS brain lesions from human subjects, Lamp2 and LC3II/I ratios are decreased, suggesting impaired autophagy [211]. Further studies have shown serum and CSF concentrations of Atg5 are elevated in MS patients. This study also noted increased expression of Parkin in both serum and CSF with higher serum levels of lactate [212]. In addition, blood from MS subjects show altered expression of several autophagy-related genes (these included *ATG9A, BCL2, FAS, GAA, HGS, PIK3R1, RAB24, RGS19, ULK1, FOXO1,* and *HTT*) [213]. Autophagy and mitophagy are imperative for immune cell function, differentiation, and adaptive immunity. The role of these pathways in driving autoreactive T cell differentiation in MS needs to be understood.

### 3.5. Amyotrophic Lateral Sclerosis

ALS is a neurodegenerative disease marked by the loss of alpha motor neurons in the lumbar spinal cord and motor cortex [214,215]. The lifespan post ALS diagnosis is short, often 2–3 years, because progressive muscle wasting leads to lung paralysis. In some cases of ALS, dementia is present [216]. Most ALS cases are sporadic, with a rare subset (less than 5% of total cases) being familial. These familial cases are caused by mutations in genes including *Tdp43*, *Fus*, *Fig4*, *Ang*, *Vapb*, and *C9orf72* [217,218,219,220]. Mutations in the *Optn* gene which encodes an autophagy protein, optineurin, were found to be causative of ALS in 2010 [221,222,223]. After the discovery of mutations in *Sod1* and *Tdp43*, transgenic mouse models were developed [224,225,226].

Changes to mitochondrial ultrastructure in human ALS subjects were revealed several decades ago [227,228,229]. Cytoplasmic inclusions that may represent mitochondria-containing autophagic vacuoles are observed in ALS motor neurons [230,231]. Although ALS neurodegeneration is anatomically specific, mitochondrial abnormalities are found systemically [214,215,227,229,232,233]. Mitochondrial dysfunction is present in platelet and muscle mitochondria from ALS subjects [227,229,234,235,236,237,238]. mtDNA may contribute to ALS pathologies, as cybrid cells harboring mtDNA from ALS subjects often show mitochondrial abnormalities and increased cell death [150,229,239,240].

ALS is modeled using rodents that express mutant *SOD1* or mutant *TDP43* [224,226]. SOD1 is a cytoplasmic enzyme which was identified within mitochondrial membranes [225,229,241,242,243]. This is the case for both mutant SOD1 and to a lesser extent wild type SOD1. Mutant *SOD1* ALS transgenic mice have altered mitochondrial morphology and mitochondrial SOD1 accumulation [242,243]. This raises the possibility that mutant SOD1 may drive neurodegeneration by damaging mitochondria. TDP43 mutants are also observed within mitochondria and appear to induce mitochondrial dysfunction [242,243,244,245,246]. Both TDP43 and SOD1 are known to aggregate within motor neurons and muscle; TDP43 interacts with proteins critical to mitophagy in an inhibitory manner [219,244,245,247,248,249,250,251].

Impaired mitophagy was proposed to be involved in the denervation of neuromuscular junctions in an ALS mouse model [252]. Lysosomal dysfunction has also been implicated in ALS. Specifically, lysosomal deficits result in an abnormal accumulation of autophagic vacuoles that engulf damaged mitochondria within the motor neuron axons of *G93A SOD1* ALS mice [177]. Impaired mitochondrial turnover along with the accumulation of misfolded proteins and protein aggregates contributes to ALS-linked mitochondrial dysfunction and motor neuron death. Mitochondrial and mitophagy ultrastructure are varied across compartments of motor neurons [253,254]. Parkin, Miro1, and Mfn2 are depleted in an ALS mouse model (*G93A SOD1*); however, mitochondrial localized p62 is upregulated [255]. Mutant forms of optineurin interfere with Parkin ubiquitin ligase function [256].

In human post-mortem samples, increased autophagic vesicles are observed in lumbar spinal cord motor neurons [257]. Induced pluripotent stem cell (iPSC)-derived motor neurons from familial ALS subjects with *C9orf72* mutations or haploinsufficiency have dysfunction of autophagy pathways [258,259].

As discussed above mitochondrial dysfunction and mitophagy alterations are prevalent in human AD, PD, MS, and ALS samples as well as cell and animal models of disease. We discuss below the methods being investigated to modulate mitophagy.

## 4. Modulating Mitophagy in Neurodegeneration

Increasing mitophagy in transgenic mouse models of neurodegeneration have shown mostly beneficial effects. In AD models (iPSC derived, transgenic mice, and *C. elegans*), increasing mitophagy using nicotinamide mononucleotide (NMN), UA, or actinonin (AC) reduced Aβ and tau aggregation. In AD transgenic mice, mitophagy induction benefited cognition [260,261]. These compounds are NAD^+^ precursors, which may drive mitophagy through alterations in redox balance (NAD^+^/NADH). UA likely drives mitophagy through a PINK1/Parkin/Nix axis.

Broad autophagy induction with Rilmenidine in the *G93ASOD1* mouse model of ALS did not change disease progression [262]. The mechanism(s) of Rilmenidine autophagy/mitophagy induction are currently unknown. Rapamaycin (an mTOR inhibitor) treatment of this same mouse model was detrimental unless mature lymphocytes were depleted [263]. These studies highlight the importance of understanding the non-cell autonomous effects of autophagy and mitophagy pathways.

In PD rodent models (MPTP injection), a drug, Salidroside, increased Parkin and PINK1 expression and preserved dopaminergic neurons in the substantia nigra [264]. A cell permeable form of Parkin rescued cells from aggregating α-synuclein, partially restored motor function, and protected dopaminergic neurons in the 6-OHDA PD (6-hydroxydopamine) mouse model [265].

In an MS-related mouse model of EAE administration of rapamycin, an mTOR inhibitor improves outcomes [211]. Further studies of the EAE mouse model show that excessive activation of Drp1 through nitration leads to an overaction of mitophagy [266]. Blocking this pathway alleviated the disease burden in the EAE mouse mode [267]. Genetic ablation of *Beclin 1* was also protective in the EAE mouse model [268]. Overall, in MS inhibition of mitophagy specifically in T cells could be beneficial.

UA has been shown to be safe and well-tolerated in elderly adults, with plasma concentrations detectable at a range of doses. Furthermore, UA affected mitochondrial gene expression in muscle [269]. A separate study in healthy adults is registered for UA (NCT04160312), but no results have been posted. Clinical trials for NMN (NCT04228640 safety trial) are recruiting or ongoing (NCT03151239 effects on cardiometabolic health). In Japan, the first human clinical trial of NMN showed no deleterious effects, suggesting NMN is tolerable and safe [270,271]. No clinical trials for these NAD^+^ precursor mitophagy modulators are currently registered for neurodegenerative diseases.

Lifestyle interventions could be useful tools to boost mitophagy. Exercise and diet have been shown to induce mitophagy [272,273,274,275,276]. In both AD animal models and human clinical trials, exercise has shown cognitive benefit [275,277,278,279,280,281]. The exercise effects in ALS and PD are more controversial, but overall, exercise seems to improve physical and cognitive outcomes [282,283,284,285,286,287]. Intermittent fasting and ketogenic diets have also been shown to induce mitophagy and improve cognition/motor performance [288,289,290,291,292,293,294].

Current clinical trials aimed at increasing autophagy, mitophagy, or mitochondrial function are ongoing or recently completed. For AD, these include treatment with nicotinamide riboside (NR; NCT04430517; NAD^+^ precursor), Dimebon (NCT00675623, NCT00829374; stimulates mTOR-dependent mitophagy), resveratrol (NCT00678431; mTOR inhibitory), ketogenic diets (NCT03860792), and caloric restriction diets (NCT02460783). In PD, these include nicotinamide supplementation (NCT03568968; NAD^+^ precursor), ubiquinol/Coenzyme Q10 (NCT03061513; autophagy mechanism unknown), ketogenic diets, and ketone esters (NCT01364545, NCT04477161). In MS, clinical trials include ketogenic diet, dimethyl fumarate, and MitoQ (NCT03740295, NCT04267926, NCT02461069). In ALS, one clinical trial for ubiquinol/Coenzyme Q10 (NCT00243932; autophagy mechanism unknown) was completed. Overall, the clinical trials directly modulating mitophagy are lacking and require more attention. The majority of mitophagy inducers in clinical trials have unknown mechanisms and pleotropic affects.

Targeting specific pathways and tissues could be advantageous in avoiding deleterious or off-target effects. Designing new therapeutic strategies should focus on modulating specific mitophagy targets while also enhancing mitochondrial function and biogenesis.

## 5. Concluding Remarks

Mitophagy and mitochondrial quality control are important mechanisms which should be further studied in the context of brain aging and neurodegeneration. Novel mechanisms of mitochondrial quality control in neurons and glia have illuminated the knowledge gaps in this field of study. Mitochondria are dynamic and multifaceted organelles at the forefront of pathways associated with aging and neurodegeneration (proteostasis, metabolism, inflammation, and synapse loss). Thus, targeting mitochondrial health and mitochondrial quality control will target the most common pathological mechanisms in neurodegeneration.

## Figures and Tables

**Figure 1 ijms-21-09661-f001:**
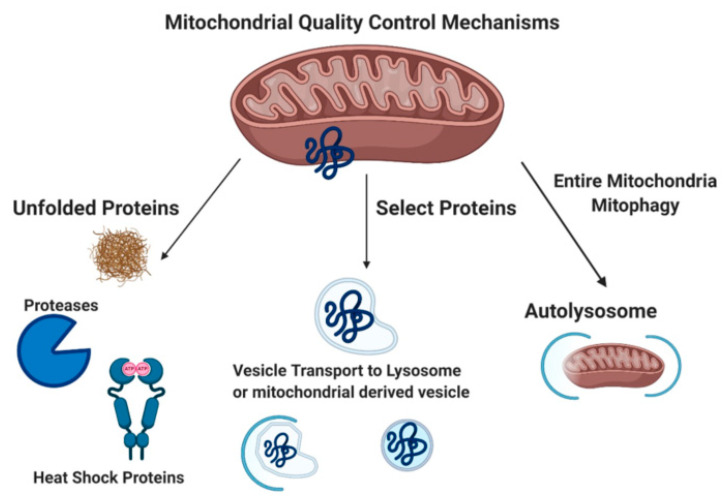
Mitochondrial Quality Control Mechanisms. Unfolded protein response, mitochondrial derived vesicles, and mitophagy [23,33] Created with BioRender.com.

**Figure 2 ijms-21-09661-f002:**
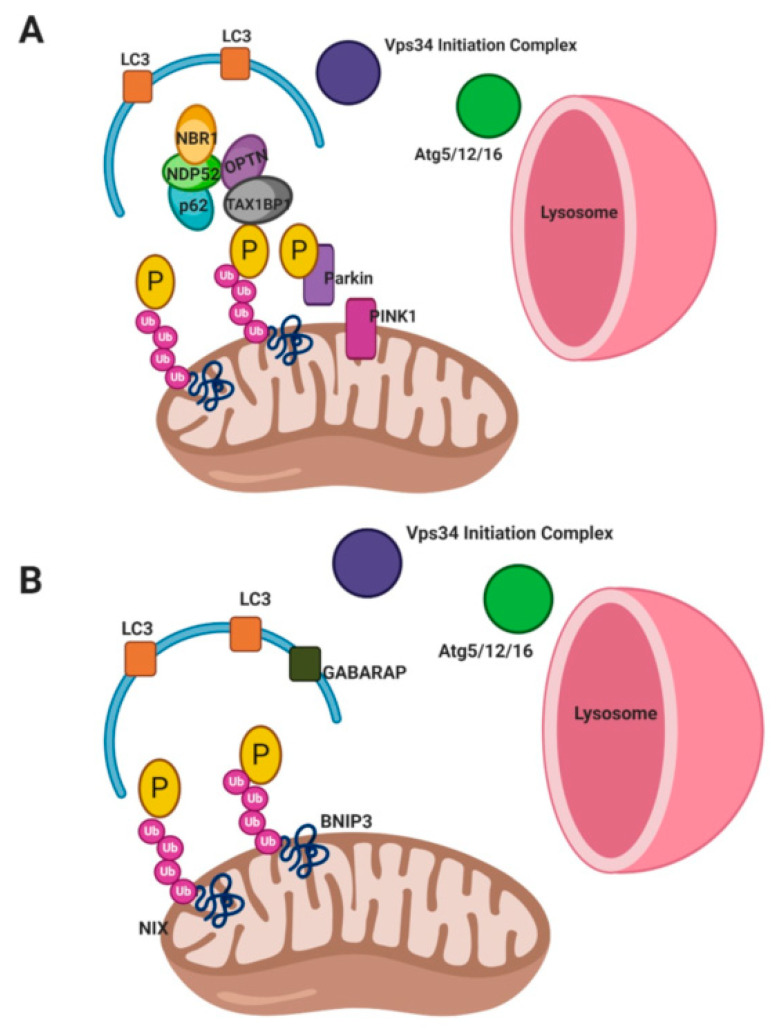
Mitophagy. (**A**) Non-Receptor Mediated Mitophagy or Classical Mitophagy. Activation of PINK1 leads to recruitment of ubiquitin and Parkin. Parkin ubiquitinates and phosphorylates mitochondrial proteins (such as VDAC, MFN1/2, and TOM20) and this initiates receptor adaptor protein recruitment (p62, NDP52, OPTN, TAX1BP1, and NBR1). These adaptor proteins interact with LC3 to form the autophagosome. The Vps34 and Atg5/12/16 complex facilitate autophagosome maturation, closure, and lysosome fusion [44,50] (**B**) Receptor Mediated Mitophagy. Mitochondrial receptor proteins (BNIP3, NIX, FUNC1, AMBRA1, PHB2, or cardiolipin) are ubiquitinated and phosphorylated. This facilitates their interaction with LC3 and GABARAP for autophagosome formation. The Vps34 and Atg5/12/16 complex facilitate autophagosome maturation, closure, and lysosome fusion [44,50]. Created with BioRender.com.

**Figure 3 ijms-21-09661-f003:**
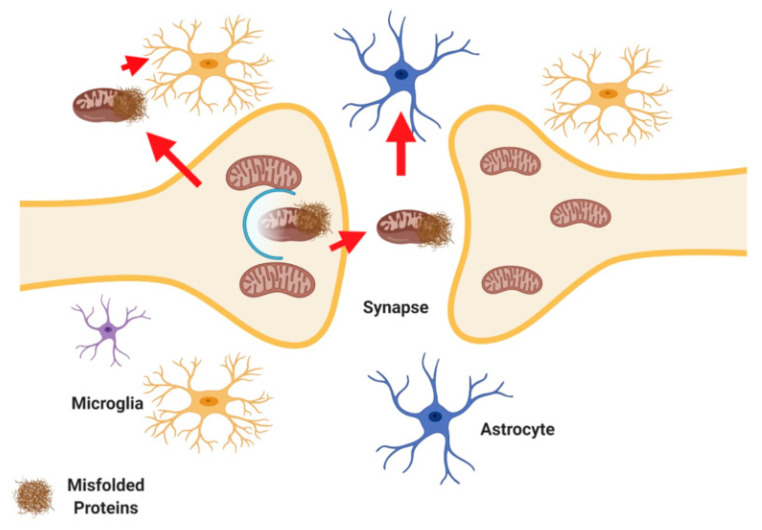
Transcellular Mitophagy. Mitochondria are released at neuronal synapses where they are taken up by glial cells (astrocytes and/or microglia) for degradation [8,9]. Created with BioRender.com.

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
