# Peer review of "Mitophagy and the Brain"

_ijms, 2020, doi:10.3390/ijms21249661_

Round 1

Reviewer 1 Report

The article entitled “Mitophagy and the Brain” by Natalie S. Swerdlow1and Heather M. Wilkins submitted for publication in the International  Journal of Molecular Sciences is coming in queue with numerous other publications in the same domain (there are almost 138 review dealing with mitophagy and brain). So the authors have to come view new idea or an original way of thinking, that is not really the case (there are 138 results coming out from Pubmed for brain and mitophagy reviews). So, for sure the authors have to offer a granted value to their own work to be published despite the fact that they are not specialists of the autophagy field nor of the mitochondrial one. I am very sorry but  whatever is the number of references cited, I think that the authors can short the number of reference by citing some reviews that punctuated the evolution of the field to be able to delate some references Galluzi L. et al. 2017,  Green DR et al. 2011 etc. At whole even if some parts of the article seems to be quite correct like the last paragraph “ Modulating mitophagy in neurodegeneration”, the rest of the text look like a catalog.

I think that the authors that are not specialist of mitophagy nor of mitochondria need to spend a bit more time about thinking and redrawing their article before presentation even if the fact they hardly tried to put together dispatched information in an unique review article

Line 26 – 36 the text presented on about mitochondria is quite now a banality and the authors should decide to take a reference… Within the huge literature.

The authors forget about treating some key points:

  • The origin of mitochondria and so, the possibility to sustain a bigger genome to go through the barrier of evolution has to be said in one or two sentence
  • The structure of mitochondria should be described as the outer membrane seem to have a major role in mitophagy.
  • Autophagy for the avoidance of cell death.
  • The question of mitophagy to cell. Death cross-talk is essential

Figure 1 –- Figure one appeared to be of bad quality drawing whatever the use of BioRender.

I would prefer a modified schematic representation extracted and modified from a specialized article that the authors should cite.

Somes notions should not be forgotten:

  • Mitochondria are the evolutionary relics of aerobic bacteria that invaded the proto-eukaryotic cell about a billion years ago. As such, they have a separate genome and provide the oxygen consumption-driven synthesis of ATP (via oxidative phosphorylation, OXPHOS). • However, as a side product of normal respiration, mitochondria produce reactive oxygen species (ROS) that must be detoxified.
  • Moreover, as they replicate, mitochondrial genomes accumulate mutations that eventually compromise the efficiency of OXPHOS. Mitochondrial deficiency, excessive ROS, or both appear to be driving forces in aging because they reduce cellular fitness, inflict damage to other organelles, or cause mutations of the nuclear genome.
  • Beyond their roles in the chronic process of cellular and organismalaging, mitochondria also mediate acute cell death. In many instances, developmental, homeostatic, and pathological cell death involves a critical step in which mitochondria release proteins that trigger the self-destructive enzymatic cascade that causes apoptosis. This point should be treated a the cross-talk between mitophagy and cell death.

Figure 2 - Even if interesting t e figure stayed of bad quality (to low definition of the lines and text). The publications that served as refence of this drawing should be cited and the acknowledgement should be here…

Figure 3 – same remarks than for Figure 22

References: 278 references is to much for such an article

Author Response

We thank the reviewer for their time and courtesy to assist us with improving our review article.

We appreciate the feedback on our review article. We have addressed the comments below. However, we kindly request the reviewer keep in mind Dr. Wilkins is within her first year of independent principle investigator status. Despite this, Dr. Wilkins has 26 articles published on mitochondria (out of over 45 total articles). We appreciate the support while Dr. Wilkins establishes her career as a young investigator.

While we were asked to reduce the number of citations by citing other reviews, we don’t feel this is necessary nor appropriate given that the second reviewer (as well as this reviewer) have asked us to add more information in certain areas. We don’t feel it serves the research community to cite reviews over original research in a review article. This is the only request we have not adhered to. If the journal requests us to reduce the number of citations, we will comply.

We addressed the following:

  1. Figures 2 and 3 are original work of the authors and original ideas of the authors. Citations discussing the theories presented are in the text as necessary and are now added to the figure legend.
  2. We have altered figure 1 and cited a reference as requested. We opted to change the figure to focus on mitochondrial quality control.
  3. All figures are high quality (300 DPI). We are certain at publication the journal will ensure the high-quality figures we submitted will be incorporated at high resolution.
  4. We added the following sections as requested:
    1. The origin of mitochondria
    2. The structure of mitochondria
    3. Autophagy and mitophagy with relation to cell death
    4. ROS and the crosstalk between mitophagy and cell death (and we’ve cited your excellent article which had an excerpt included in your review comments).

Reviewer 2 Report

The review manuscript by Swerdlow and Wilkins provides a good description regarding the role of mitophagy in aging and neurodegenerative diseases (in particular, Alzheimer’s Disease, Parkinson’s Disease, and amyotrophic lateral sclerosis).  The manuscript is of interest to the International Journal of Molecular Science readers. My comments for potential further improvement are shown hereafter:

  • What was the literature search criteria and reasoning to include these three diseases? If the results from the search is now not available, please include which search engines and MeSH terms were used.
  • For example, multiple sclerosis is now increasingly recognized as a neurodegenerative disease with an average MS patient being in the 55-65 years old, where immunosenescence, oxidative stress, and autophagy play a major pathophysiological role. Changes of increased to decreased mitophagy may exacerbate the neurodegeneration in MS. I would recommend adding an additional short section for MS as well.

Overall change in MS disease reviewed in: Vaughn CB, Jakimovski D, Kavak KS, et al. Epidemiology and treatment of multiple sclerosis in elderly populations. Nat Rev Neurol 2019;15:329-342.

Some non-extensive examples of autophagy and mitophagy related work in MS:

Patergnani S, Castellazzi M, Bonora M, et al. Autophagy and mitophagy elements are increased in body fluids of multiple sclerosis-affected individuals. J Neurol Neurosurg Psychiatry 2018;89:439-441

Igci M, Baysan M, Yigiter R, et al. Gene expression profiles of autophagy-related genes in multiple sclerosis. Gene 2016;588:38-46.

Liang P, Le W. Role of autophagy in the pathogenesis of multiple sclerosis. Neurosci Bull 2015;31:435-444.

Alirezaei M, Fox HS, Flynn CT, et al. Elevated ATG5 expression in autoimmune demyelination and multiple sclerosis. Autophagy 2009;5:152-158.

Castellazzi M, Patergnani S, Donadio M, et al. Correlation between auto/mitophagy processes and magnetic resonance imaging activity in multiple sclerosis patients. J Neuroinflammation 2019;16:131.

  • Please add a sentence in the abstract describing the process of mitophagy. This will improve the readability for a greater audience.

Author Response

We thank the reviewer for their time and courtesy to assist us with improving our review article.

We appreciate the thoughtful review and feedback. 

We addressed the following:

  1. Included the search engines and terms used.
  2. A section on MS has been added with the references provided.
  3. Added a sentence in the abstract describing the process of mitophagy.

Round 2

Reviewer 1 Report

The arguments concerning the status of teh first authors are not valuable concerning the publication of an article. Nevertheless, the authors have made consequent changes and have clearly improved the manuscript. So, I do not see any treason to not accept this manuscript.